# The Prognostic Role of Intratumoral Stromal Content in Lobular Breast Cancer

**DOI:** 10.3390/cancers14040941

**Published:** 2022-02-14

**Authors:** Carina Forsare, Sara Vistrand, Anna Ehinger, Kristina Lövgren, Lisa Rydén, Mårten Fernö, Ulrik Narbe

**Affiliations:** 1Division of Oncology, Department of Clinical Sciences Lund, Lund University, SE-223 81 Lund, Sweden; carina.forsare@med.lu.se (C.F.); anna.ehinger@med.lu.se (A.E.); kristina.lovgren@med.lu.se (K.L.); marten.ferno@med.lu.se (M.F.); 2Department of Oncology, Växjö Central Hospital, SE-352 34 Växjö, Sweden; sara.vistrand@kronoberg.se; 3Department of Clinical Genetics and Pathology, Skåne University Hospital, Lund University, SE-221 85 Lund, Sweden; 4Division of Surgery, Department of Clinical Sciences Lund, Lund University, SE-223 81 Lund, Sweden; lisa.ryden@med.lu.se; 5Department of Surgery, Skåne University Hospital, SE-214 28 Malmö, Sweden; 6Department of Oncology, Skåne University Hospital, SE-221 85 Lund, Sweden

**Keywords:** intratumoral stroma, lobular breast cancer, long-term prognosis

## Abstract

**Simple Summary:**

High intratumoral stromal content is related to worse outcomes in several types of cancer. However, its prognostic role in breast cancer seems to differ between different subtypes. High intratumoral stromal content is a negative prognostic marker in triple-negative breast cancer, while the opposite is the case for estrogen-receptor-positive breast cancer, in which higher stromal content is indicative of a better prognosis. Most lobular breast cancers are estrogen-receptor-positive, and the tumor tissue has a clearly defined histological appearance, often with a high intratumoral stromal content. To date, the prognostic role of intratumoral stromal content in lobular breast cancer remains unclear. In this study, we aimed to investigate the prognostic importance of intratumoral stromal content in estrogen-receptor-positive lobular breast cancer. Our results show that high intratumoral stromal content is an easily assessed and clinically useful indicator of a good prognosis in lobular breast cancer.

**Abstract:**

Previous studies have shown that high intratumoral stromal content is associated with a worse prognosis in breast cancer, especially in the triple-negative subtype. However, contradictory results have been reported for estrogen-receptor-positive (ER+) breast cancer, indicating that the prognostic role of intratumoral stromal content may be subtype-dependent. In this study, we investigated the importance of intratumoral stromal content for breast cancer-specific mortality (BCM) in a well-defined subgroup (*n* = 182) of ER+/human-epidermal growth-factor-receptor-2 negative (HER2−) invasive lobular breast cancer (ILC). The intratumoral stromal content was assessed on hematoxylin–eosin-stained whole sections and graded into high stroma (>50%) or low stroma (≤50%). A total of 82 (45%) patients had high-stroma tumors, and 100 (55%) had low-stroma tumors. High-stroma tumors were associated with a lower Nottingham histological grade, low Ki67, and a luminal A-like subtype. After a 10-year follow-up, the patients with high-stroma tumors had a lower BCM (HR: 0.43, 95% CI: 0.21–0.89, *p* = 0.023) in univariable analysis. Essentially the same effect was found in both the multivariable analysis (10-year follow-up) and univariable analysis (25-year follow-up), but these findings were not strictly significant. In ER+/HER2− ILC, high intratumoral stromal content is an easily assessable histological indicator of a good prognosis.

## 1. Introduction

Invasive lobular cancer (ILC) is the second most common type of breast cancer after invasive ductal cancer of no special type (NST) and constitutes 10 to 15% of all breast cancer [1]. Compared to NST, ILC is characterized by lower cellular content and displays a different and characteristic infiltrative growth pattern, where the cancer cells typically grow in thin lines (i.e., ‘single file’) into the surrounding tissue [2,3]. The intratumoral stromal content is higher in ILC compared to tubular breast cancer and NST [4]. Moreover, several differences regarding the tumor microenvironment in the stroma have been identified between ILC and NST [5]. Studies have shown that stromal cells (e.g., pericytes and fibroblasts) integrate with cancer cells and can affect tumor progression via paracrine signaling, which stimulates tumor growth and angiogenesis [6]. Tumor infiltrating lymphocytes (TILs) in the intratumoral stroma interact with cancer cells via complex immune-mediated mechanisms, and the presence of TILs is associated with prognostic outcome in breast cancer [7,8]. To date, there is no strict consensus on how to assess the relationship between stromal content and cancer cells in tumor tissue. Most commonly, a manual assessment of the percentage of stromal content in relation to tumor cellularity in a representative area of the tumor tissue using hematoxylin–eosin-stained sections is applied, and a cut-off value of 50% intratumoral stromal content is used as a threshold for grading tumors as ‘high stroma’ or ‘low stroma’ [9,10,11]. An alternative method of grading is the tumor stroma ratio (TSR), where the epithelial area is divided by the stromal area [4]. The intratumoral stromal content has also been evaluated using digital image analysis and machine learning algorithms [4].

More than 10 years ago, Mesker et al. demonstrated that intratumoral stromal content is an independent prognostic factor for colon cancer [12]. The study showed that tumors with high stromal content are correlated with a worse prognosis in terms of both overall and disease-free survival. Since then, further studies have reported that high intratumoral stromal content corresponds to a worse outcome in various other tumor types (e.g., non-small cell lung cancer, hepatocellular carcinoma, and esophagus cancer) [13]. In line with these results, several studies on breast cancer have shown that high intratumoral stromal content is a negative prognostic factor, especially in triple-negative breast cancer [10,14,15,16,17,18,19]. However, in studies investigating estrogen-receptor-positive (ER+) breast cancer, the opposite pattern was indicated, namely that higher stromal content is associated with a better prognosis [4,11]. The prognostic role of intratumoral stromal content can, therefore, differ greatly between different subtypes of breast cancer. The relevance of considering different subtypes was also indicated in one study where there was no evidence for the prognostic importance of intratumoral stromal content in inflammatory breast cancer [20]. Most previous studies have included all the histological subtypes of breast cancer and, hence, only a small fraction correspond to ILCs. To our best knowledge, there are no previous studies investigating the prognostic role of intratumoral stromal content exclusively in ILC. The vast majority of ILCs are classified as estrogen-receptor-positive and human-epidermal growth-factor-receptor-2 negative (ER+/HER2−). The unique infiltrative growth patterns of ILC, with typically high stromal content in relation to low tumor cellularity, make the study of this cancer as an individual entity even more relevant.

The aim of this study was, therefore, to investigate the prognostic role of intratumoral stromal content in relation to breast cancer-specific mortality (BCM) in a well-defined subgroup of ER+/HER2− ILC. Furthermore, the correlations between intratumoral stromal content and other clinicopathological variables were studied.

## 2. Materials and Methods

### 2.1. Study Population

The patients included in the present study originated from a cohort of female breast cancer patients diagnosed with ILC at the Departments of Pathology, Skåne University Hospital Lund, and Helsingborg Hospital, Sweden between 1980 and 1991 (*n* = 276; Figure 1) [21,22]. The inclusion criteria were as follows: (1) histologically verified ILC cases (re-evaluated by two breast pathologists), (2) available whole sections for stromal assessment, (3) available tissue microarray (TMA) core for immunohistochemical analyses, and (4) presence of the ER+/HER2− subtype. In total, 182 cases were included in the study (Table 1).

Information on Ki67 and progesterone receptor (PR) status were obtained from TMA [22]. The Nottingham histological grade (NHG) was evaluated on whole sections according to Elston and Ellis [23]. A further histological subdivision of the ILCs was not performed. E-cadherin was characterized using immunohistochemistry (IHC) (Clone NCH-38, M3612 DAKO/Agilent 1:100), and loss of E-cadherin was confirmed in approximately 85% of the included tumors. Clinicopathological characteristics and follow-up data were retrieved from medical charts and pathology reports. In the present study, 83 patients (46%) received adjuvant endocrine treatment and 5 (3%) received adjuvant chemotherapy. None of the patients received neoadjuvant therapy. At the end of the study, 54 patients (30%) died from breast cancer and 76 (42%) from other causes. The remaining 52 patients (28%) still alive in June 2015 had a median follow-up time of 26 years (range 0.7–35 years) [21,22]. Using the same classification as described in our previous report [22], which is based on the 2017 St. Gallen surrogate definition of the intrinsic subtypes, this cohort of ER+/HER2− tumors was divided into luminal A-like (NHG 1+2, low Ki67, and progesterone receptor (PR) > 20%) and luminal B-like (at least one of the following three criteria fulfilled: NHG 3, high Ki67, or PR ≤ 20%) [24].

### 2.2. Immunohistochemical Assays of PR and Ki67

The expression of PR (Clone PgR636, M3569 DAKO 1:100) was analyzed using IHC on TMA, and a score of ≥1% stained nuclei was considered positive. The expression of PR was also analyzed with a cut-off value of 20% for the luminal-like classification. The proliferation marker Ki67 (Clone MIB-1, M7240 DAKO 1:200) was analyzed using IHC on TMA and considered high if ≥24% cells were considered positive. The cut-off value was set at this level to mimic the fraction of high Ki67 tumors in our previous whole section analyses of ILC [21]. TMA preparation was carried out as previously described in Ref. [22].

### 2.3. Assessment of Intratumoral Stromal Content

Using bright-field microscopy, evaluation of the intratumoral stromal content was conducted on whole hematoxylin–eosin-stained tissue sections (4 µm) by two investigators (S.V. and K.L.) blinded to all tumor, patient, and follow-up information. The same microscope was used during the whole study (magnification 100×). A visual estimation of the whole section was made, and the area with the highest tumor cellularity was selected. The selected area was then evaluated and graded as high stroma (>50% of the area consisted of intratumoral stroma) or low stroma (≤50% of the area consisted of intratumoral stroma; Figure 2) in line with the cut-offs used in several other previous studies [10,11,14,19]. In case of disagreement between the two evaluators, a joint review was performed to reach a consensus.

### 2.4. Statistical Analysis

The statistical analyses were carried out using the software Stata version 16 (StataCorp, College Station, TX, USA). Contingency tables and χ^2^ tests were used to correlate the stromal content with lymph node status, tumor size, NHG, PR, and Ki67. Cox regression (proportional hazards model) with competing risk was used after a follow-up time of both 10 and 25 years to compare the BCM between the high stroma and low stroma groups. Multivariable analysis and survival were then calculated using the Cox regression model.

## 3. Results

### 3.1. Intratumoral Stromal Content and Association with Other Clinicopathological Variables

Eighty-two of the 182 patients (45%) had high-stroma tumors, and 100 patients (55%) had low-stroma tumors (Figure 2; Table 1). High intratumoral stromal content was strongly associated with lower NHG (*p* < 0.001), and no NHG 3 tumors were in the high-stroma group (Table 1). High intratumoral stromal content was also associated with low Ki67 (*p* = 0.049) and luminal A-like tumors (*p* = 0.009) but not with the other clinicopathological characteristics (Table 1).

### 3.2. Breast Cancer Mortality

After a 10-year follow-up, intratumoral stromal content (high vs. low) was associated with a lower BCM (HR: 0.43, 95% CI: 0.21–0.89, *p* = 0.023; Figure 3; Table 2). In the multivariable analysis adjusted for NHG, Ki67, tumor size, lymph node status, age, and adjuvant endocrine treatment, the evidence for an independent effect of tumor stroma content was weaker (HR: 0.45, 95% CI: 0.16–1.26, *p* = 0.128; Table 2).

After a 25-year follow-up, the evidence of an association between intratumoral stromal content and BCM was less pronounced in both the univariable (HR: 0.61, 95% CI: 0.35–1.07, *p* = 0.083) and multivariable analyses (HR: 0.73, 95% CI: 0.34–1.57, *p* = 0.417).

### 3.3. Exploratory Analyses

Essentially the same results were found in the subgroup analyses investigating the prognostic effect of intratumoral stromal content in relation to BCM in patients with tumors classified as luminal A-like (*n* = 111; 10-year follow-up—HR: 0.45, 95% CI: 0.16–1.33, *p* = 0.150; 25-year follow-up—HR: 0.55, 95% CI: 0.25–1.20, *p* = 0.130; Figure 4A), luminal B-like (*n* = 63; 10-year follow-up—HR: 0.43, 95% CI: 0.12–1.50, *p* = 0.186; 25-year follow-up—HR: 0.75, 95% CI: 0.29–1.92, *p* = 0.546; Figure 4B), and NHG2 (*n* = 145; 10-year follow-up—HR: 0.50, 95% CI: 0.22–1.13, *p* = 0.094; 25-year follow-up—HR: 0.65, 95% CI: 0.34–1.24, *p* = 0.191; Figure 4C). High intratumoral stromal content was associated with an improved prognosis (Figure 4A–C), although the evidence in these numerically smaller subgroups was weaker than the evidence seen in the whole cohort.

## 4. Discussion

In this study of a well-defined population of patients with ER+/HER2− ILC, high intratumoral stromal content was found to be a ‘good-prognostic’ factor associated with lower BCM based on a follow-up time of 10 years. The evidence was weaker when the follow-up was prolonged to 25 years and in multivariable analyses with 10- and 25-year follow-ups. High intratumoral stromal content was closely associated with NHG, which, together with Ki67, nodal status, and patient age, comprised the most important prognostic factors in the multivariable analysis. Interestingly, none of the 13 NHG 3 tumors presented high intratumoral stromal content, which was in stark contrast to the large proportion of high-stroma tumors among NHG 1 tumors (19 of 22).

These results are in line with those of two previous reports on ER+ breast cancer showing that patients with high-stroma tumors have a better prognosis than those with low-stroma tumors [4,11]. Additionally, in one of these studies, the same strong association between high intratumoral stromal content and low NHG was reported [4]. However, the prognostic importance of intratumoral stromal content in ILC was not specifically addressed in these studies and the number of included patients with ILC was low (i.e., in Millar et al., 31 out of 403 and, in Downey et al., 6 out of 180). In studies that included all histological subtypes or exclusively analyzed triple-negative breast cancer, the patients with high-stroma tumors were repeatedly associated with a worse prognosis [10,14,15,16,17,18,19]. The discrepancy in the results obtained in different studies suggests that the prognostic role of intratumoral stromal content may differ in different subgroups. The aim of the present study was, therefore, to specifically study ER+/HER2− ILC, which is a well-defined subgroup including most patients with ILC. In exploratory analyses, the prognostic effects of the intratumoral stromal content in patients with tumors classified as luminal A-like, luminal B-like, and NHG 2 were essentially the same as those in the whole cohort. In line with previous ILC studies [25,26], the majority (81%) of the patients in this cohort were classified as NHG 2, an intermediate prognostic group for whom long-term outcomes are unpredictable and, therefore, complementary prognostic tools are warranted. In our subgroup analysis, patients with NHG 2 ILC, the patients could be subdivided into two comparable groups of high vs. low intratumoral stromal content (high stroma: *n* = 62; low stroma: *n* = 83) with different outcomes (Figure 4C).

A recent study by Vangangelt et al. showed that intratumoral stromal content increases with age, and that this content is not an independent factor in multivariable analysis for patients above or at the age of 70 years [27]. In the present study, only three breast-cancer-related deaths were recorded in this older age group. Consequently, age dependency could not be further examined. Furthermore, adjuvant systemic therapies vary between different studies. In a study by Roeke et al., the association between high intratumoral stromal content and an impaired outcome was most distinct in the patients treated with adjuvant endocrine therapy [9]. Although the evaluation of intratumoral stromal content seems to be easily assessable, with satisfactory kappa values reported (0.68–0.85; [9,10,14]), differences in the evaluations of intratumoral stromal content on hematoxylin–eosin-stained sections may still be critical (e.g., the area selected, the size of the selected area, and the number of assessed areas) [10,11,28,29]. Digital image analysis and machine learning algorithms may provide a more reproducible approach than manual scoring [4]. Further studies analyzing the method agreement and prognostic validity between manual and digital assessment in the same set of tumor samples would be of interest. 

Although prognostic information can be derived from assessment of the intratumoral stroma, it is important to consider that this is a simplistic way of looking at the highly complex underlying tumor microenvironment. The importance of distinguishing between different dominant stromal components (collagen, fibroblasts, or lymphocytes) and combining the stromal content in the primary tumor with that in lymph-node metastases has also been indicated [30,31]. Differences between the ILC and NST microenvironments regarding cancer-associated fibroblasts and intratumoral vessels have been observed [5]. In ILC, the expression of fibroblast-activation protein alpha and fibroblast-specific protein 1/S100A4 was higher in both the tumor and stromal cells than in NST [32]. In tubular breast cancer and NST, CD34+ fibroblasts are absent in the intratumoral stroma [33,34], whereas, in ILC, these fibroblasts are present in approximately two thirds of the cases [35]. A study by Westhoff et al. showed that a loss of CD34+ fibroblasts in ILC was associated with an unfavorable prognosis [35].

The prognostic significance of TILs in TNBC and HER2+ breast cancer has been extensively studied, and the presence of TILs in the intratumoral stroma is associated with a more favorable prognosis [7,8]. In ER+/HER2− breast cancer, a subtype normally less immunogenic than TNBC and HER+ breast cancer, the prognostic role of TILs is not fully elucidated, although several studies suggest that the presence of TILs is associated with a less favorable prognosis [7]. Furthermore, in ILC, only a small fraction of the tumors have TILs, and the levels are generally lower than in NST [36,37]. The prognostic role of TILs in ILC has hitherto been sparsely investigated; however, two recent studies indicate that the presence of TILs is an unfavorable prognostic factor in this histological subtype [37,38]. In a study by Millar et al., patients with TNBC and a combination of low intratumoral stromal content and low levels of TILs had an extraordinarily poor prognosis [4]. In the present study, no assessment of TILs was performed. A large and sufficiently powered study investigating the prognostic impact between the intratumoral stromal content and TILs in ILC would be encouraged. 

In line with our results, a protein-based stromal signature was used to identify a special good prognostic subtype of breast cancer called ‘reactive’. Members of this subtype are mainly ER+/HER2−, either lobular or tubular, and feature a high number of stromal cells [39]. The possible predictive value of intratumoral stroma was indicated in one study using gene-expression data, showing that increased stromal gene expression can predict resistance to preoperative chemotherapy with 5-fluorouracil, epirubicin, and cyclophosphamide [40]. Finally, in a recent publication, reactive stroma was found to be associated with trastuzumab resistance in HER2-positive primary breast cancer [41].

The re-evaluation of exclusively pure ER+/HER2− ILCs by breast pathologists is a strength of the present study. Another important advantage is this study’s long-term follow-up since the lobular histological subtype is associated with a risk of late recurrences that can occur >10 years after the initial primary diagnosis. The patients with ILC in this cohort were, to a great extent, not treated with any systemic adjuvant therapy, which afforded an opportunity to study the prognostic importance of the intratumoral stroma in a state of only minor interference with the original tumor biology. A limitation of the present study is the relatively small sample size, which is associated with a low power. Future studies exploring the clinical importance of intratumoral stromal content in ILC are warranted and should preferably be performed using larger and well-defined patient materials.

## 5. Conclusions

The present study showed that, in patients with ILC, high-stroma tumors are associated with a better clinical outcome than low-stroma tumors. In ER+ breast cancer, intratumoral stromal content appears to have the same prognostic role in ILC as other histological subtypes. Ultimately, high intratumoral stromal content is an easily assessable histological indicator of a good prognosis and can potentially complement established clinical prognostic factors.

## Figures and Tables

**Figure 1 cancers-14-00941-f001:**
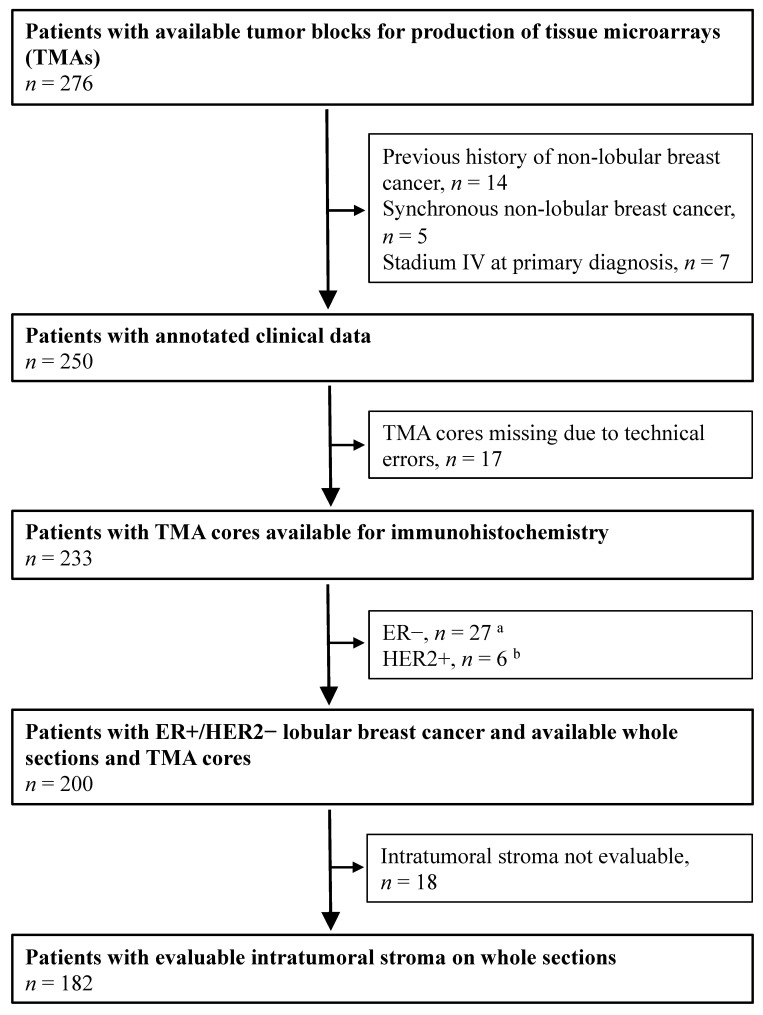
Flowchart of female breast cancer patients with tumors primarily classified as lobular breast cancer at the Departments of Pathology, Skåne University Hospital Lund, and Helsingborg Hospital (1980–1991). ^a^ Estrogen-receptor (ER) expression (1D5, DAKO, 1:100) was analyzed through immunohistochemistry (IHC) of whole sections. ER-positivity (+) was defined as >10% stained nuclei. ^b^ Human-epidermal growth-factor-receptor-2 (HER2) expression (CB11, Novocastra, 1:200) was analyzed using IHC on tissue microarray (TMA) and categorized into 4 different IHC groups depending on the cell membrane staining intensity: 0, 1+, 2+, 3+. A value of 3+ was considered positive (HER2+). A HER2 amplification test was not performed.

**Figure 2 cancers-14-00941-f002:**
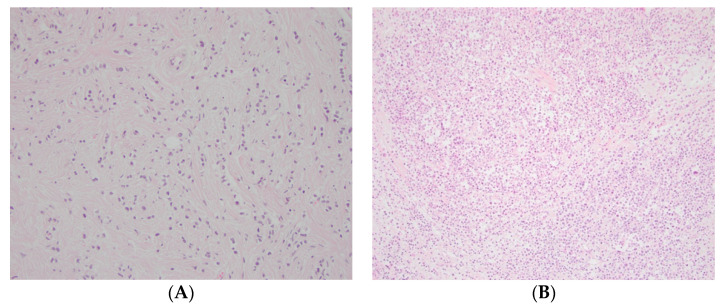
Representative photos of (**A**) high and (**B**) low intratumoral stromal content on hematoxylin–eosin-stained whole sections (magnification 100×).

**Figure 3 cancers-14-00941-f003:**
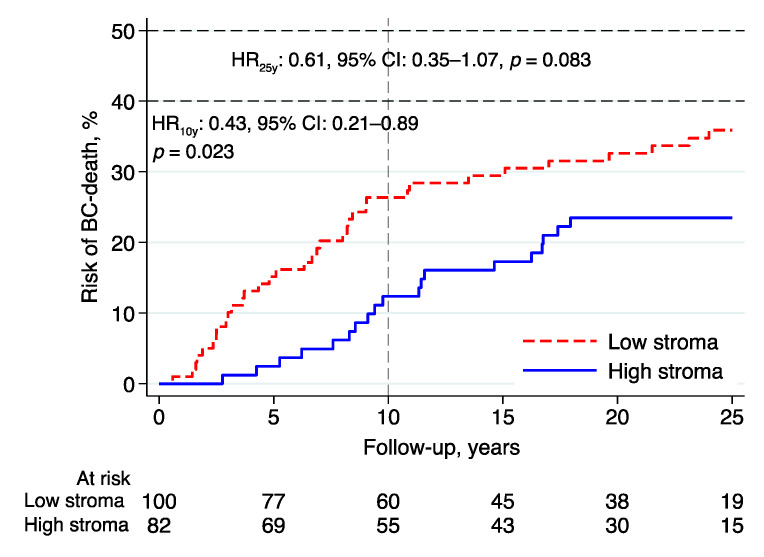
Risk of breast cancer death in lobular breast cancer by intratumoral stromal content; 10- and 25-year follow-up. Abbreviations: BC—breast cancer; CI—confidence interval; HR—hazard ratio.

**Figure 4 cancers-14-00941-f004:**
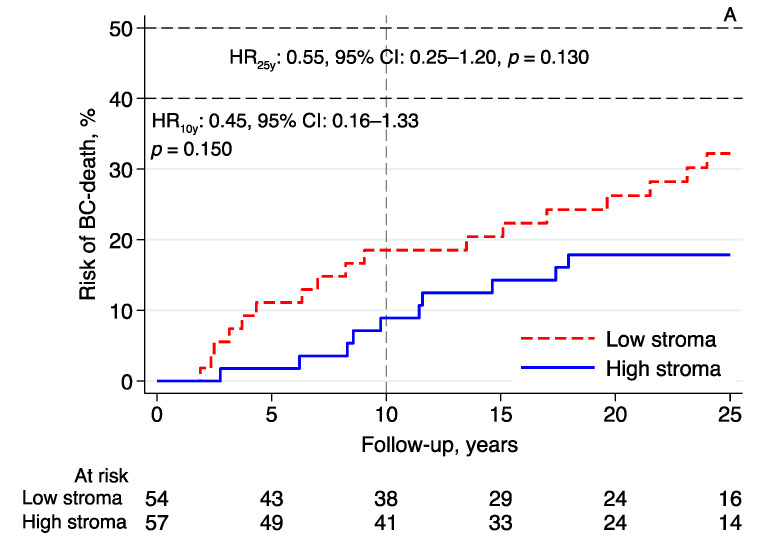
Risk of breast cancer death in lobular breast cancer by intratumoral stromal content; 10- and 25-year follow-up for (**A**) luminal A-like, (**B**) luminal B-like, and (**C**) NHG 2. Abbreviations: BC—breast cancer; CI—confidence interval; HR—hazard ratio; NHG—Nottingham histological grade; stroma; intratumoral stromal content.

**Table 1 cancers-14-00941-t001:** Patient and tumor characteristics in relation to intratumoral stromal content.

Variables	*n* = 182	High Stroma *(n* = 82)	Low Stroma *(n* = 100)	*p*-Value
Age, median (range)	63 (39–85)	63 (37–85)	62 (40–86)	0.581 ^a^
Menopause				0.326 ^b^
Pre	43	17	26	
Post	133	64	69	
Missing	6			
Size				0.281 ^b^
0–20 mm	103	49	54	
>20mm	76	30	46	
Missing	3			
Nodal status				0.359 ^b^
0	107	54	53	
1–3	27	11	16	
4+	39	15	24	
Missing	9			
NHG				<0.001 ^b^
1	22	19	3	
2	145	62	83	
3	13	0	13	
Missing	2			
PR				0.827 ^b^
<1%	35	16	19	
≥1%	142	62	80	
Missing	5			
≤20%	46	19	27	0.661 ^b^
>20%	131	59	72	
Missing	5			
Ki67				0.049 ^b^
Low (<24%)	161	74	87	
High (≥24%)	12	2	10	
Missing	9			
Luminal				0.012 ^b^
Luminal A-like	111	57	54	
Luminal B-like	63	20	43	
Missing	8			
Surgery				0.568 ^b^
BCS	43	21	22	
Mastectomy	139	61	78	
Endocrine treatment ^c^				0.159 ^b^
Yes	72	28	44	
No	109	54	55	
Missing	1			
Chemotherapy ^c^				0.503 ^b^
Yes	5	3	2	
No	176	79	97	
Missing	1			
Recurrence ^d^				0.018 ^b^
Yes	75	26	49	
No	107	56	51	

Abbreviations: NHG—Nottingham histological grade; PR—progesterone receptor; Ki67—proliferation marker; luminal A-like—NHG 1+2, low Ki67, and PR > 20%; luminal B-like—NHG 3, high Ki67, or PR ≤ 20% (at least one of the three criteria fulfilled); BCS—breast conserving surgery. ^a^
*p*-value from Mann–Whitney U-test; ^b^
*p*-value from Pearson’s chi-square test; ^c^ adjuvant treatment; ^d^ including: local, regional, and distant recurrence.

**Table 2 cancers-14-00941-t002:** Patient and tumor characteristics associated with breast cancer mortality after 10-year follow-up in univariable and multivariable analyses.

Variables	*n*	Univariable 10 Years			Multivariable (*n* = 157) 10 Years		
		HR	95% CI	*p*-Value	HR	95% CI	*p*-Value
Age (years)	182	0.94	0.91–0.97	<0.001	0.93	0.89–0.98	0.002
Size (0–20 vs. >20 mm)	179	3.56	1.78–7.14	<0.001	1.64	0.64–4.21	0.304
Nodal status							
N0	107	1.00			1.00		
N1 (1–3+)	27	1.66	0.58–4.71	0.342	0.67	0.17–2.68	0.566
N ≥ 2 (4+)	39	5.92	2.85–12.33	<0.001	2.83	1.02–7.80	0.045
NHG (3grps)							
1	22	0.22	0.03–1.61	0.136	1.05	0.12–9.55	0.967
2	145	1.00			1.00		
3	13	3.93	1.51–10.21	0.005	5.68	1.81–17.86	0.003
PR (≤1 vs. >1%)	177	0.64	0.30–1.37	0.248	0.24	0.08–0.72	0.010
Ki67 (high vs. low)	173	5.33	2.30–12.36	<0.001	4.00	1.41–11.33	0.009
Stroma (high vs. low)	182	0.43	0.21–0.89	0.023	0.45	0.16–1.26	0.128
ET (yes vs. no)	181	0.52	0.27–1.00	0.048	0.58	0.22–1.52	0.270

Abbreviations: HR—hazard ratio; NHG—Nottingham histological grade; PR—progesterone receptor; Ki67—proliferation marker; stroma—intratumoral stromal content; ET—endocrine treatment.

## Data Availability

The datasets used and/or analyzed in the present study are available from the corresponding author upon reasonable request.

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
