# Peer review of "The Prognostic Role of Intratumoral Stromal Content in Lobular Breast Cancer"

_cancers, 2022, doi:10.3390/cancers14040941_

Round 1

Reviewer 1 Report

In this manuscript by Forsare et al., the authors investigate the prognostic value of intratumoral stromal content in lobular ER+/HER2- breast cancer. This article is of interest to clinicians since it could bring a new tool to predict the outcome of this particular subset of patients. Although the results obtained are interesting, the lack of statistical significance in the multivariate analysis represents an issue. As suggested by the authors, this may be due to the relatively small sample size, which is a limitation of the present study. This reviewer encourages the authors to include a higher number of patients in the study, to see if the statistical significance in the multivariate analysis can be achieved.

As indicated by the authors, digital image analysis may provide a more reproducible approach than manual scoring to evaluate intratumoral stromal content. This reviewer encourages the authors to use a digital image method to explore the stromal content and compare the results obtained with the results from manual scoring to see if the prognostic value could be better assessed using another method.

Author Response

Thank you for reviewing our manuscript.

Please see the attachment for our replies to all comments in a point-by-point way. The comments are highlighted in blue, the replies are written in italics and all changes have been marked by yellow in the new version of the manuscript.

Reviewer 2 Report

The intratumoral stroma content is an important feature for defining the diagnosis in breast cancer patients. The Authors considered a group of patients with infiltrating lobular carcinoma and highlight the correlation between high levels of infiltrating component and favorable prognosis with a follow-up of 10 years. The proposal to consider this factor as a prognostic index is original in patients with NHC grade 1-2 infiltrating lobular carcinoma and may have a positive clinical impact.

The major limitation of the study is determined by the small number of patients considered in the study. Which suggestion could be useful to evaluate the opportunity to associate this feature with the use of genomic tests that could better guide the adjuvant therapeutic strategy and provide a predictive value.

Author Response

(The authors gave the same response as above.)

Reviewer 3 Report

Forsare et al. provide a concise retrospective cohort study comparing breast cancer mortality between patients with ER+/HER2- invasive lobular carcinoma with >50% intratumoral stroma to those with </= 50% intratumoral stroma. They found that high stroma (>50%) was associated with lower breast cancer mortality at 10 years on univariate analysis, however this association did not remain statistically significant on multivariate analysis as high intratumoral stroma also correlated with other prognostic markers including tumor grade and Ki67. They have a very clear hypothesis and provide an organized, well written manuscript. My thoughts for improvement are below.

Specific Criticisms:

  1. In the introduction, would it be possible to emphasize prior studies (such as Nakagawa et al, ref #29) demonstrate a difference in stromal content between ILC and IDC and that frequently referenced study Downey et. al. which showed prognostic significance of stromal content in luminal tumors only included 6/180 cases of ILC? This may help in emphasizing how the present study adds to the current body of literature.

  1. Results (section 3.1 & Table 1):

- What percent of patients were female vs. male?

- What percent of patients received anti-estrogen therapy & adjuvant chemotherapy between the high & low stromal groups? This would be important to know when comparing mortality (and anti-estrogen therapy in multi-variate analysis).

- Was there a difference in rate of mastectomy vs. lumpectomy + radiation?

  1. The authors note that on multi-variate analysis, the association between stromal content and mortality is lost due to other confounding variables. What then is the utility of assessing stromal content if other variables which are currently assessed routinely (such as grade, Ki67) have a stronger association with mortality? Is it possible for the authors to create a scoring system using stromal content as well as these other variables? And if so, does this create a better model compared to each variable by itself?

  1. The data show that stromal content failed to reach statistical significance in multivariate analysis, suggesting that high vs. low stromal content was not independently predictive of survival. The same was true for subgroup analyses. In the analysis, stromal content was used as a categorical variable (high >50% vs. low <50%). Are the data for multivariate analysis significant if stromal content was analyzed as a continuous variable (reported as percent stromal content from 0% to 100%).  

  1. Is there a different “cut-point” for high vs. low stromal content where the data related to effect on survival are more robust. Maybe 50% is not the right choice.

  1. Was there any difference in stromal content and disease recurrence (rather than just mortality)?

  1. Other reports have shown that CD34+ stomal expression is predictive of outcome in invasive lobular breast cancer with stromal loss of CD34+ fibroblasts significantly associated with lower overall and disease-free survival rates (Westhoff CC et al. Virchows Arch. 2020; 477:717-724). It would be helpful data to stain your TMA with CD34 and refine the findings with these data.

  1. In the discussion, are the authors able to provide their thoughts on how assessment of stromal content would improve upon patient care and/or prognostication of outcomes as compared to what is currently used? How will the results of their study change our current practices?

Author Response

(The authors gave the same response as above.)

Round 2

Reviewer 1 Report

The responses addressed the issues in my previous review. All of my concerns have been addressed.